# Implementation of Novel Quality Assurance Program for Hepatitis C Viral Load Point of Care Testing

**DOI:** 10.3390/v14091929

**Published:** 2022-08-30

**Authors:** Wayne Dimech, Liza Cabuang, Kylie Davies, Giuseppe Vincini

**Affiliations:** National Serology Reference Laboratory, Melbourne 3065, Australia

**Keywords:** point of care testing, quality assurance, hepatitis C virus, pilot

## Abstract

All patients should have access to accurate and timely test results. The introduction of point of care testing (PoCT) for infectious diseases has facilitated access to those unable to access traditional laboratory-based medical testing, including those living in remote and regional locations, or individuals who are marginalized or incarcerated individuals. In many countries, laboratory testing for infectious diseases, such as hepatitis C virus (HCV), is performed in a highly regulated environment. However, this is not the case for PoCT, where testing is performed by non-laboratory staff and quality controls are often lacking. An assessment of the provision of laboratory-based quality assurance to PoCT for infectious disease was conducted and the barriers to participation identified. A novel approach to providing quality assurance to PoCT sites, in particular those testing for HCV, was designed and piloted. This novel approach incudes identifying and validating sample types that are inactivated and stable at ambient temperature, creating cost-effective supply chains to facilitate logistics of samples, and the development of a smart phone-enabled portal for data entry and analyses. The creation and validation of this approach to quality assurance of PoCT removes the barriers to participation and acts to improve the quality and accuracy of testing, reduce errors and waste, and improve patient outcomes.

## 1. Introduction

Laboratory-based testing for hepatitis C virus (HCV) is well established and the quality of results are assured through a range of regulatory requirements applied to test kit providers and laboratories [1,2,3,4]. In Australia and most other countries with a stringent regulatory authority, HCV test kits are classified as high risk, requiring the company providing the kits to show evidence of quality, safety and performance [5]. Laboratories testing for HCV are required to implement strict quality management systems, including documented staff training and competency, standard operating procedures (SOP), equipment calibration and maintenance, as well as participating in accredited quality assurance (QA) procedures such as external quality assessment schemes (EQAS) and quality control (QC) programs [1,6]. EQAS consist of programs where participants are sent a panel of positive and negative samples for testing. The participant tests the samples blinded and reports the results to the EQAS provider. Results for each test site are analyzed for accuracy and reported back to the participant. QC programs consist of participants testing known positive and negative samples frequently and monitoring the variation in test results over time [7]. For many years, point of care testing (PoCT) for HCV and other high-risk pathogens have been used to provide access to test results while the patient is present at the clinic, allowing for the initiation of treatment and counselling if the results are positive. PoCT is now well established in both developed and low-middle income countries (LMIC) [8,9]. However, application of QA of PoCT lags behind laboratory-based testing, where arguably there is a greater need [10,11,12,13,14].

All patients should have access to accurate and timely test results. Some of the most disadvantaged populations, including first nation peoples; those living with stigma; the poor and socially disadvantaged; remote and regional populations; and many at-risk populations have limited or no timely access to laboratory-based testing [8,15]. False negative test results cause a lost opportunity for treatment and the on-going spread of infection, whereas false positive results can lead to unnecessary treatment and anxiety. Poor testing leads to waste, medicolegal issues, a loss of confidence in the tests and clinics and, ultimately, poor patient outcomes [10,12,13,14,16]. For these reasons, the World Health Organization (WHO) encourages all testing be performed under a quality assured environment [17]. Well-designed QA programs are critical to reducing inaccurate PoCT for HCV and other infectious diseases.

### 1.1. Laboratory Testing for HCV

The diagnosis and management of HCV infections is achieved through the detection of antibodies to HCV (serology) and the detection and quantification of viral nucleic acids circulating in venous or peripheral blood through nucleic acid testing (NAT) [18]. In advanced laboratory settings, both serology and NAT are performed using sophisticated, expensive, high-throughput instrumentation. Often, laboratories test serum or plasma for anti-HCV antibodies using a screening assay and confirm any initially reactive samples using a second, different assay before reporting positive results. Generally, binding assays using a chemiluminescent signal are used. NAT viral load testing uses a reverse transcriptase polymerase chain reaction (RT-PCR), calibrated against a WHO international standard, with results reported in international units per milliliter (IU/mL). All tests used in a laboratory accredited to ISO 15189 must be validated prior to use, staff must be trained and demonstrate ongoing competence, instruments must be routinely calibrated and maintained, and SOPs must be followed [1]. The performance of the assays is monitored using QC sample testing, with the results plotted on a Levey–Jennings chart [7,19]. Any unexpected deviation is investigated and, if found to indicate a risk of false patient results, testing is suspended until a root cause of the variation is identified and rectified. It is a requirement of ISO 15189 that laboratories assess the entire testing process using an EQAS [1]. Laboratories are audited by an independent third party periodically to ensure compliance with the relevant standard(s). The staff performing the tests are laboratory professionals with tertiary qualifications in laboratory medicine or similar, and evidence of their competency and training are de rigueur. The testing is overseen by a clinical pathologist who is responsible for the communication of abnormal results to the treating physician, usually reviewing results of other blood-borne virus and sexually transmitted infection testing performed on the patient to provide a whole-of-health assessment.

### 1.2. PoCT for HCV

Both serology and NAT are available for PoCT for HCV [20]. Serology testing is most commonly performed by testing finger-stick peripheral blood using a single-use lateral flows device (RDT). A drop of blood is added to a specimen well. On addition of a manufacturer-supplied buffer, the blood migrates through a nitrocellulose strip until it meets bound HCV antigens, being the test line. If the sample contains anti-HCV antibodies a reaction appears on the test line. The RDT usually has a control line as well as the reaction line. A reaction will appear on the control line if the patient’s sample migrates to that point, indicating that flow of sample was adequate. No reaction at the control line indicates an invalid test, which must be repeated. The results are read subjectively, usually by non-laboratory personnel, and results are transcribed manually onto worksheets and/or delivered verbally to the patient. Most RDTs do not provide positive and negative controls as separate components, so test sites in a low prevalence setting may not experience many (if any) reactive results.

Molecular testing for HCV RNA at point of care is achieved by a cartridge-based assay, testing a single sample at a time [8]. The current assays available on the market are enclosed technologies where serum or finger-stick whole blood is applied to a cartridge using a manufacturer-supplied applicator. The cartridge, which contains all the required components for testing, is introduced into a dedicated instrument for processing and a result is reported. The instruments usually have a range of inbuilt process validation. An internal control (e.g., the detection of human-derived material), indicates that a sample has been introduced into the cartridge. A volume control is also commonly used, indicating that sufficient volume of patient sample was added. An error will be reported if the internal or volume controls are invalid, and no result will be issued. The instruments also have a range of electronic controls to monitor temperature, sample flow, and other critical processes throughout the test. However, none of these controls monitor the accuracy of the results reported.

In many PoCT settings, the training and knowledge of the individual performing the test varies considerably, from well-developed training programs and ongoing competency assessment to minimal training or competency assessment. Manual testing such as RDT is prone to traceability issues [21]. If the processes used to test patients and record results are not well established and followed, sample mix-up and transcription errors can be experienced, leading to the incorrect results being reported. Reaction lines can be faint and the reading of RDTs subjective, creating the potential of misreading or misinterpreting the results. A standard process for reading and interpreting the results is required, or false results can be reported. Testing with cartridge-based NAT is less open to error, but traceability and transcription errors remain, as the results often are recorded manually. It is important, therefore, to monitor the performance of HCV PoCT using a QA program designed for PoCT.

At the time of writing, there were only four RDTs, four enzyme immunoassays and four NAT assays for HCV testing prequalified by WHO [20]. The QA of PoCT in general, and for HCV in particular, varies between different settings. Although some intervention programs using PoCT for infectious diseases implement QA programs to monitor the quality of testing, this approach is not universally applied. Where QA programs are provided, they are based on laboratory QA programs, but there are multiple barriers to providing laboratory-based QA programs to PoCT settings. These barriers have been detailed elsewhere [22]. Briefly, they can be summarised as

**Sample types**—Whereas laboratory-based testing uses serum or plasma, PoCT often use peripheral whole blood for both HCV serology and NAT. QA programs should use the sample matrix similar to the patient’s sample tested by the participant.**Sample logistics**—Sending samples to testing sites poses some difficult barriers. The samples used in laboratory-based QA are often biological materials that require being shipped either chilled or on dry ice that complies with the international air transport association (IATA) standard. An accredited EQAS provider is responsible for the delivery of the samples to the participant and ensures QA samples are received in good condition. Shipping is expensive and access to remote and regional sites are extremely difficult. Servicing numerous PoCT sites with multiple but small sized packages is not cost effective and administratively cumbersome. Therefore, the number of shipments per year should be kept to a minimum.**Lack of infrastructure**—Many PoCT sites lack the infrastructure commonly found in laboratories. Laboratory-based QA samples usually require refrigerated or frozen storage. The addition of samples into the testing devices commonly require pipettes or utilise manufacture-provided applicators. Therefore, the samples provided must be suitable for a range of different test kits. Some sites in very remote geographies or in places experiencing conflict or natural disasters do not have access to electricity and/or pure water.**Fixed test events**—Usually, EQAS providers conduct several test events per year, where the EQAS samples are sent to the test sites just prior to the opening of the test event and the test site has a defined period of time to test and report the results. After the deadline, the EQAS provider analyses the results submitted by all participants and issues a report. This process is partly to reduce collusion (where test sites compare results before submission). Set test events create a barrier to PoCT QA, because the QA samples must be received by the site at a particular time, exacerbating the logistics issues. Additionally, not all PoCT sites perform testing continuously. Some PoCT sites are mobile. Others may have no tests (or expired kits) available at the time of the test event. The test events may also be scheduled during significant festival/holiday time, times of conflict or civil unrest, monsoon, or other extreme weather conditions when delivery of samples is disrupted. The site, therefore, is unable to participate, wasting the cost of the programs and the shipping.**Cost**—Many PoCT sites that are funded to participate in EQAS use international laboratory-based programs, which are costly both in subscription rates and shipping costs. Once the external funding ceases, so does the participation in the QA programs.**Regulations**—Quality control samples are in-vitro diagnostic devices (IVDs) subject to internationally applied conformity assessment. They must be approved for use by the local regulatory authority before they are supplied to the market. This requirement makes international QC samples costly. Often there is no access to locally produced materials. There are few QC samples that have been developed specifically for PoCTs.**Data collection and management**—Even when PoCT sites participate in a QA program, the data are often not collected and stored in a central database, and rarely are the data used to monitor the performance of the test kit, in particular for qualitative tests. Whereas errors in reporting of specific sample testing are investigated, data obtained from many test sites using the same sample set can expose unexpected trends that may indicate poor IVD performance. A lack of centrally stored and analyzed QA data is a lost opportunity to monitor the quality of PoCTs in the field.**Data analysis**—The analysis and reporting of quantitative QA data is well-defined. Traditionally, the quantitative results of QC programs are plotted on a Levey–Jennings chart and acceptance limits are applied. Although there is some debate on how those limits should be calculated, there is a general acceptance that this process should be used. Where possible, it is ideal that test sites using the same assay and QC compare their results by using a peer-to-peer QC monitoring system. National Serology Reference Laboratory, Australia (NRL) provides an international QC program called QConnect™ that allows peer comparison. However, there is a poorer understanding of how to manage qualitative data such as those derived from RDT testing. The use of a rating system to note the intensity of the bands can convert qualitative data from nominal to ordinal, allowing more sophisticated analysis.

### 1.3. Novel Approach to the Monitoring of HCV Molecular PoCT

NRL is a WHO Collaborating Centre whose mission is to promote the quality of testing for infectious diseases, globally and has provided internationally accredited QC [7] and EQAS for laboratories testing for infectious diseases for decades. Over the past five years, NRL has participated in several programs providing QA services to PoCT settings in collaboration with WHO, Foundation for Innovative and New Diagnostics (FIND), Flinders University International Centre for Point-of-Care Testing and Kirby Institute Viral Hepatitis Clinical Research Program. These activities have provided NRL the opportunity to develop novel PoCT QA programs for a range of infectious disease analytes, including HCV testing. These novel PoCT QA programs overcome the barriers identified above and supports the National Health and Medical Research Council Partnership Project Grant for the National HCV Point-of-Care Testing Program.

## 2. Materials and Methods

The NRL developed a PoCT QA program to monitor HCV viral load testing, comprised of several components each of which required validation or implementation. Each component is designed to overcome the barriers to accessing quality assurance by sites performing PoCT for HCV RNA.

**Quality assurance programs offered:** An approach devised by NRL and WHO to monitor PoCT uses several different QA challenges [22]. To assess the competency of the operator after training and throughout the period of testing, a “Competency panel” comprised of a known positive and a negative sample is provided. The positive dried tube sample consists of HCV genotype 1 at a viral load between 3 and 4 log10 IU/mL on reconstitution. These samples are tested by the operator after theory training and before patient testing to demonstrate competence, and then periodically over time. At a minimum, NRL recommends testing one vial per month (alternatively one positive and one negative each other month, i.e., six Competency panels per year), but optimally it is recommended that one Competency panel (i.e., one positive and negative vial) should be tested each week that testing is performed. The results of testing are submitted to NRL via a graphical user interface of EDCNet™ (https://edcnet.nrlquality.org.au, accessed on 17 August 2022), an internet-based software designed to monitor the performance of testing of infectious diseases.

In addition to Competency panels, each test site should participate in an EQAS. NRL EQAS format includes five vials of dried tube sample along with reconstitution buffer. Each positive sample can contain different HCV genotypes and different viral loads within different challenges. NRL designs each EQAS challenge to include the range of HCV genotypes to reflect genetic variation encountered globally. Under the National HCV program, test sites are provided two test events per year. The samples are tested blinded, and results reported to NRL for analysis. In 2022, NRL is using OASYS (Oneworld Accuracy, Vancouver, BC, Canada) as the informatics system to collect and analyse the EQAS results. In 2023, the data collection and analysis will be performed by EDCNet™, using a mobile phone portal described below. NRL recommends that test sites perform at least two EQAS challenges per year.

**Sample type:** NRL uses a dried tube sample format for the novel HCV PoCT molecular QA programs. Dried tube samples were first described by USA CDC [23,24]. NRL has since modified and validated the method for use in HCV RNA testing of plasma [25]. More recently this approach has been expanded to include a whole blood format, where aliquots of packed, washed human red blood cells are resuspended in infected human plasma, having a known viral load. The resultant material is air-dried in a plastic screw-top tube and stored at −80 °C until use. A decrease in viral load from the initial levels is experienced during drying, but once dried, the samples have been validated as stable and homogeneous, a requirement of ISO 17043 accreditation.

To validate the stability of the HCV whole blood dried tube sample format, multiple vials of samples comprising of HCV genotype 1 and genotype 3, each at a high and low vial load (Table 1) were manufactured and stored at −20 °C (Table 1). To assess real-time stability, one vial of each format was retrieved from storage and tested on the Cepheid GeneXpert HCV FS assay (Cepheid, Sunnyvale, CA, USA) each month for a period of 15 months.

In parallel, to assess accelerated stability and to mimic transport integrity and storage at various temperatures, multiple vials of samples were manufactured consisting of HCV genotype 1 at a viral load of approximately 1 × 10^4^ International units per milliliter (IU/mL). Vials were stored at 2–8 °C, 21–25 °C (room temperature), and 35 °C. Two vials stored at each temperature were removed from storage and tested every month for a period of six months. For vials stored at room temperature, stability testing was continued for a further three months. At each time point, the vials were tested in duplicate and the result in IU/mL compared.

**Sample logistics:** As described above, one of the greatest challenges to providing QA programs for PoCT is the logistics of shipping QA panel samples to PoCT sites. To overcome this barrier for the National HCV program, NRL collaborates with the test kit manufacturer Cepheid to provide the Competency and EQAS panels with the test reagents, thereby eliminating the cost of distribution and assuring the panels are shipped with identical conditions to the test reagents.


**Data collection and management:**
**Competency panel results:** The systematic collection of not only QA test results but also associated metadata is critical to a QA program. Metadata include the date of testing, operator identification, test kit name and lot number, sample identification, test result and, where appropriate, any comments relevant to the testing performed. Therefore, the data collected and the mode of submission must be simple.**EQAS panel results:** EQAS are usually organized so that all participants test the same samples at the same time. This is partly due to reduce collusion, where participants discuss the results prior to submission. An EQAS designed to allow participants to test ad hoc must overcome this situation. The samples in the NRL PoCT EQAS have a four-digit, alpha-numeric code. Each EQAS sample type is manufactured in bulk. For example, 1000 vials of HCV genotype 3 high viral load, being the “true result”, will be manufactured. NRL has designed a database whereby the samples within this manufacturing lot are labelled with unique four-digit codes (i.e., 1000 different four-digit codes) but each code is related back to the “true result” of that manufacture lot. The five-member EQAS panels are comprised of different sample types selected randomly. Therefore, on receipt of the EQAS panels, the participant will be unable to identify the true results of the EQAS samples, as the samples within their panels will be different to those received by others. On data entry of the result of an EQAS sample, the four-digit code is compared with the “true result” in the database. The participant will receive notification of the accuracy of the results immediately on submission.


**Data collection:** In LMICs, access to a fixed, standalone, internet-enabled computer in PoCT sites is not universal. However, there is general access to internet-enabled mobile phone technology. To facilitate easy data collection, NRL aimed to design, build, and validate a mobile phone portal that uses QR code technology. All competency and EQAS panel boxes contain a QR code. This code contains the identification of the panel and associated sample vials, and panel and sample lot numbers. Once a site is enrolled into the program via the phone, the mobile phone retains information regarding the site code, so that site details, such as test kits used, can be retrieved from the database.

**Data analysis:** All results of competency and EQAS testing are submitted and stored in a centralized database of EDCNet™, along with metadata such as test kit lot numbers, operator identification, panel/vial lot numbers and, where relevant, the instrument serial number. These data allow NRL staff to perform data analysis to detect shifts and trends over time. Acceptance limits, using pre-established QConnect™ Limits processes are applied to quantitative data, whereas the collection of ordinal data for qualitative data can allow the detection of deterioration of test kits over time. Collecting and enumerating the numbers of invalid test kits also supports the identification of substandard kit production. NRL staff review data submitted into EDCNet, using pre-established exception criteria, and investigate unexpected results in collaboration with the testing site and the test kit manufacturer.

**Cost:** A significant barrier to participation in QA by PoCT sites is the cost of international programs. The novel approach described above reduces this barrier considerably. The large-scale manufacture of QA samples that can be used across Competency and EQAS, using small amounts of infected plasma, reduces the cost of production. This technology can also be transferred to local manufacturing sites, using locally acquired samples. Delivering EQAS and Competency panels in bulk to centralized distribution hubs and the use of existing supply chains to deliver the QA panels to the test sites along with the reagents minimize the cost of shipping. The use of inactivated samples decreases the administrative costs of importation and international shipping regulation compliance. No additional cost of software development or licensing is required. The proposed model is designed to deploy the cheapest approach to achieving an effective QA program compliant with international standards and professional guidelines.

It is difficult to provide exact details of the costs for EQAS participation as there are many variables. A single program for a traditional EQAS using frozen liquid samples costs around USD 200–800, without shipping costs. The cost of shipping biological samples internationally on dry ice is often several thousands of US dollars (USD) per shipping, whereas ambient, non-biological shipments are measured in hundreds of USD. Provision of the NRL’s five-member EQAS currently costs approximately USD 75 per challenge. Increased uptake of the program is expected to decrease the cost to participants.

## 3. Results

### 3.1. Sample Type

#### 3.1.1. Real-Time Stability Testing

Dried whole blood vials comprising of HCV genotype 1 and 3, each at two different viral loads, were stored at −20 °C for a period of nine months and tested monthly. HCV genotype 3 samples were tested for a further six months (fifteen months in total). Results were plotted over time to identify any decrease in reactivity over time. Representative plot of results for HCV genotype 1 and 3, with an initial viral load of 476 IU/mL and 3020 IU/mL, respectively, are presented in Figure 1. The mean and range of results for HCV Genotype 1 and 3 were 505.3 IU/mL (range 240–617 IU/mL) and 2205.7 IU/mL (range 3020 to 1730 IU/mL), respectively.

#### 3.1.2. Accelerated Stability Testing

HCV genotype 1 whole blood dried tube samples, stored at each of 2–8 °C, 21–25 °C (room temperature), and 35 °C, were tested in duplicate monthly for a period of six months. Additional vials stored at room temperature were tested for a further three months. A total of 12 test results were analyzed for storage at 2–8 °C and 35 °C, and 18 results for storage at room temperature. The results of testing were analyzed using a box and whisker graph (Figure 2). Using ANOVA (AnalyseIT for Excel; Leeds, UK) to compare the test results obtained from each temperature showed no statistically significant difference (*p*-value = 0.668).

### 3.2. Data Collection

In collaboration with The Ashvins Group (Ashvins Group, Miami, FL, USA), NRL developed and validated a mobile phone enabled portal for the collection, storage and analysis of results from Competency and EQAS panel testing. The program, built within NRL’s EDCNet is designed for use with all infectious disease PoCT, including those reporting qualitative and quantitative results, for serology and molecular testing. A ‘screengrab’ for the data entry portal for the HCV Whole Blood Competency panel results is presented in Figure 3.

On scanning the QR code on the QA box, a data entry screen is automatically presented to the participant. The participant selects the QA sample tested and the date of testing from a menu and enters the lot number of the reagents used and the test result. Test results for NAT are quantitative and reported in the unit of measure expressed by the test system e.g., IU/mL or Cycle threshold (Ct) value. Qualitative data are entered as ordinal data (negative, 1+, 2+, 3+) or similar. Where required, a comment field is provided. On submission of the results, the participant is immediately notified whether the result submitted was correct.

## 4. Discussion

All individuals should have access to accurate and timely test results. PoCT can be used to achieve patient access; however, false test results from PoCT can result in the ongoing spread of disease, inappropriate or deferred treatment, loss of confidence in testing systems, and, ultimately, poor patient outcomes and waste of resources [10,12,13,14,16]. To avoid this situation, well-designed, fit-for-purpose QA programs must be implemented and delivered by organizations accredited to international standards. Whether in developed health systems or in LMIC, only a minority of PoCT sites testing for infectious diseases regularly participate in QA programs. Where this does occur, they tend to access laboratory-based QA programs. However, a review of laboratory-based QA programs has demonstrated that there are barriers to their implementation [22]. NRL has developed and deployed a novel approach to PoCT QA and applied it to HCV testing, as well as other analytes such as SARS-CoV-2 antigen testing.

The NRL PoCT QA program consists of sample types more relevant to PoCT, using inactivated samples stable at ambient temperatures of extended periods. The program assesses both testing competency and ongoing performance using an EQAS. The PoCT QA program utilizes existing supply chains to access remote and regional testing in geographically difficult sites using a distribution hub system. This approach reduces the cost and complexity of logistics. NRL has designed and deployed novel mobile phone technology, using QR codes on the QA panel and vials, to collect testing data and associated metadata, storing the data in a centralized database. This allows NRL staff to review and analyze data over time, with the goal of identifying and investigating unexpected results, and supporting corrective actions with the participant.

This PoCT QA approach was devised and documented through a collaboration between NRL, WHO Incidents and Substandard/Falsified Medical Products Team from the Regulation and Prequalification Department (Geneva, Switzerland) and FIND. It has since been applied to the Australian National HCV Point-of-Care Testing Program, overseen by the Kirby Institute and Flinders University International Centre for Point-of-Care Testing, and funded by the Australian government. More recently, a similar approach has been applied to a SARS-CoV-2 antigen project in 10 countries, supported by FIND. NRL seeks to expand this program to all significant PoCT analytes for serology and NAT and to partner with international bodies to transfer the technology to LMICs so that access to PoCT QA can become more universal.

Over the past five years, NRL has validated and optimized the use of dried tube and swab samples for use in QA programs [25]. The drying process inactivates the virus so that shipping as a biological is not required. After manufacture, the dried tube samples can be stored frozen at less than 20 °C for extended periods of time, allowing NRL to manufacture large numbers of vials at a time, thereby reducing the cost of production. Prior to shipping, the dried tube sample vials are packed with a known amount of a validated reconstitution buffer. The panels containing samples and buffer are shipped and stored ambient for periods of up to 6 months. At the test site, the dried sample is reconstituted in the buffer and mixed gently to resuspended. A pipette is provided to introduce the prepared sample into the test cartridge and testing proceeds as would a patient sample.

It is recommended that a “spoke and hub” system for distribution is used for PoCT panel logistics, where the QA provider delivers bulk Competency and EQAS panels to a local distributor, and the distributor ships the QA panels to the test sites along with the test kits and consumables. The local distributor could be an NGO implementing partner, ministry of health or the test kit manufacturer.

Use of mobile phone technology and QR codes with simple, intuitive data entry allows the easy of data collection, utilizing technology that is commonplace in LMICs. A well-designed, auto-populating data entry form reduces the possibility of data entry errors. Immediate analysis and reporting alerts the tester to any aberrant results. Each test site can review their results over time and compare their results to other sites using the same QA samples and test kit. All data are submitted to a central database, allowing centralized review by NRL. By accumulating the results in a central database, trends in testing can be monitored by the QA provider and the root cause of unexpected results can be determined. Where required, ministries of health, funders and purchaser of reagents, and regulators can assess the results of QA testing. Periodic reports of test quality can be developed, and unexpected assay or site performance can be investigated.

## 5. Conclusions

The WHO recommends all testing for infectious diseases be quality assured [17]. However, although PoCT has become commonplace in both developed and LMICs, to date there have been no affordable, fit-for-purpose QA programs for PoCTs. By identifying the barriers to participation in QA by PoCT sites, and redesigning programs and technologies, NRL has developed and piloted programs that can be used to monitor PoCT testing globally.

## Figures and Tables

**Figure 1 viruses-14-01929-f001:**
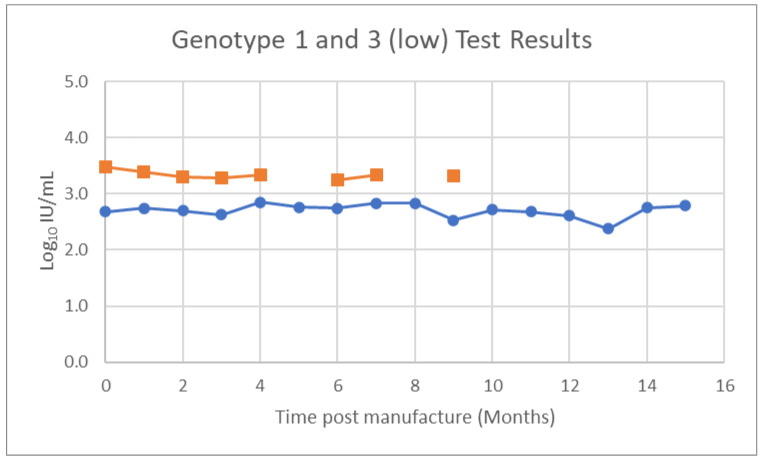
Results of monthly testing of whole blood dried tube samples for HCV Genotype 3 (blue) stored at −20 °C for 15 months, and Genotype 1 (orange) stored for nine months. Invalid results were obtained for Genotype 1 for the 5th and 8th month.

**Figure 2 viruses-14-01929-f002:**
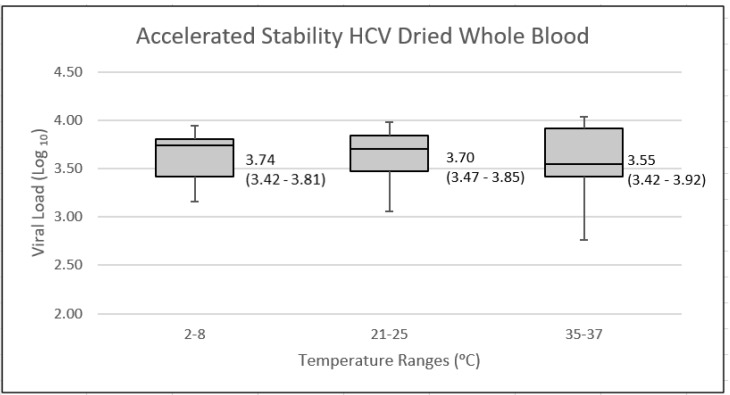
Box and whisker graph of HCV viral load results of dried tube samples store at various temperatures over a six-to-nine-month period. Media (quartile 1 and quartile 3) results, expressed in log_10_ IU/mL are presented.

**Figure 3 viruses-14-01929-f003:**
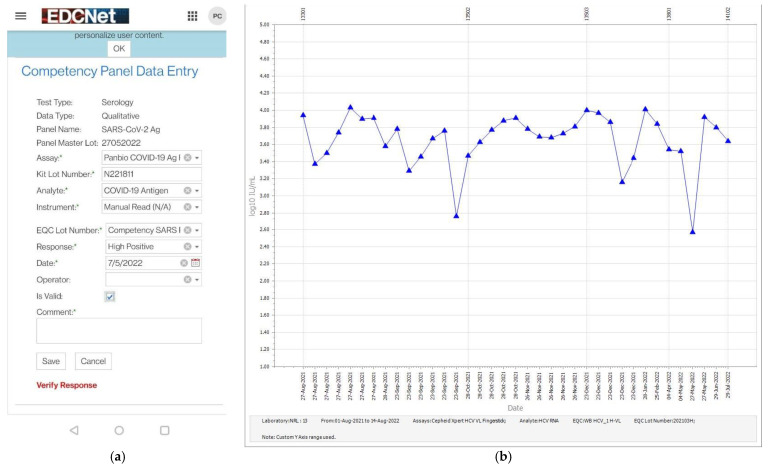
An image of the NRL Competency Panel mobile phone enabled result entry form (**a**) and representative Levey–Jennings Chart representing normal variation over time. *X* axis indicates the date of testing and *Y* axis indicates the Log_10_ IU/mL result. The change of reagent batch is displayed at the top of the graph. No acceptance limits are represented but would be displayed by horizonal bars across the graph at the upper and lower ranges (**b**).

**Table 1 viruses-14-01929-t001:** Viral load of HCV RNA whole blood dried tube samples used to determine accelerated and real-time stability for use in quality assurance programs.

Viral Load Units	HCV Genotype 1		HCV Genotype 3	
	Low	High	Low	High
Viral load (IU/mL)	3020	11,300	476	26,900
Viral load (Log_10_ IU/mL)	3.48	4.05	2.68	4.43

## Data Availability

Not applicable.

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
