# Peer review of "Implementation of Novel Quality Assurance Program for Hepatitis C Viral Load Point of Care Testing"

_viruses, 2022, doi:10.3390/v14091929_

Round 1

Reviewer 1 Report

This manuscript provides an interesting summary of efforts to provide an appropriate system for quality assurance for sites performing PoCTs. The approaches taken and the results generated should be of interest to a wide audience. As a general comment, I think the results section could be expanded as indicated below.

Specific comments:

“The RDT usually has a test line as well as the reaction line. A reaction line will appear on the test line if the patient’s sample migrates to that point, indicating that flow of sample was adequate. No reaction at the control line indicates an invalid test, which must be repeated”

So are there 3 lines – reaction line, test line and control line? I’m a bit confused as to which is which.

Fig 1 shows impressive stability of the low positive geno 3 sample. Could the authors give the mean value and range for this (ideally in IU/ml as opposed to logs) and the other 3 dried tube samples as well as this graph?

Fig 2 – could the authors indicate the extent of the whiskers in this plot ie what % variability do they cover?

Fig 3 – some interpretation of Levey-Jennings plots might be helpful. What are the criteria by which exceptional results are identified? In the example given, are those dips in apparent viral load acceptable or not?

Minor comments

There are a few typos scattered throughout the manuscript e.g.

“Some are mobile, or others may have no tests (or expired kits) available have the time of the test event”.

At total of 12 test results”

“initiative data entry allows the easy of data collection,”

“procurers and regulators can have assessed to the results of QA testing”

There are several grammatical inconsistencies eg plural nouns but singular verbs and vice-versa. Some examples:

“These novel PoCT QA programs overcomes”

“No additional costs of software development or licensing is required”

“whether the results submitted was correct.” –  all have plural nouns, singular verbs.

“Each of the components are”

“HCV genotype 1 whole blood dried tube sample, stored at each of 2-8áµ’C, 21-25áµ’C (room temperature) and 35áµ’C were tested”

“and the distributor ship” – all have singular nouns, plural verbs.

“having a known viral load are dried in a plastic screw-top tube.” This should be “having a known viral load, and are dried in a plastic screw-top tube.”

Bottom of page 5 – should “1 x 104” be 1 x 104 ?

Should “Finders University” be Flinders University?

“Thereby reducing the cost of production.” Isn’t a real sentence. Should be “, thereby …..”

Reviewer 2 Report

Summary

·        The proposal of a novel QA program for PoCT is very relevant and has the potential of high impact at public health level.

·        The authors comprehensively describe the current situation, globally, with respect to QA of PoCT and describe the challenges to providing QA very well.

·        As cost-effectiveness is one of these challenges, and the authors claim to reduce cost with their novel QA program, I feel that the authors should provide examples of estimated cost (in $US) for both local and international participants. This would enable readers to make comparisons with their own standards/practise.

·        I think the authors should provide more detail on the different genotypes provided in their samples as part of EQAS testing. If the proposed QA program is to be accessible to multiple countries around the world, then there should be some consideration given to the diverse nature of HCV around the world (8 known genotypes and over 100 subtypes). Has there been any testing on control samples of less well characterised genotypes such as genotype 7 or 8?

·        Overall, this is a very relevant paper – the background has been summarised very well, the proposed QA program is simple and therefore accessible, and the whole blood dried tube samples seem stable over a long period of time and at higher temperatures.

·        There are some minor corrections to text and figures that are highlighted in the ‘specific comments’ section below.

Specific Comments

Page 1, Abstract

·        Line 5 – there seems to be a word missing at the end of the sentence, “…highly regulated tertiary.” Are the authors referring to tertiary laboratories/hospitals etc?

·        Line 7 – I would suggest removing the sentence “When introduced to PoCT, laboratory-based quality assurance is employed. “ The sentence doesn’t add anything to the abstract.

Page 2, Introduction, Laboratory Testing for HCV

·        Line 18 – this is the first time the abbreviation EQA I used so please expand it. EQAS has been used early on in the introduction and has been expanded, but EQA is being used for the first time here.

Page 3, Introduction, PoCT for HCV, Paragraph 4

·        As of 04/08/2022 there are 4 RDTs, 4 EIA’s and 4 NAT assays for HCV testing prequalified by WHO – please update this.

Page 4, Introduction, PoCT for HCV, Bullet point titled ‘Fixed test events’

·        Line 7 – the sentence reads “have no tests (or expired kits) available have the time of the test event.” Please replace the second ‘have’ with ‘at’.

Page 4, Introduction, Novel Approach to the Monitoring of HCV Molecular PoCT

·        Line 10 – NHMRC abbreviation is being used for the first time, please expand it.

Page 4 , Materials and Methods title

·        Suggest moving the title, ‘Materials and Methods’ to the top of page 5

Page 5, Materials and Methods

·        Line 2 – the word ‘of’ is missing in the sentence, “several components each ___ which required validation or implementation.”

Page 5, Materials and Methods, 2.1 Quality assurance programs offered, Paragraph 1

·        Line 11 – please put in the webpage reference after EDCNet. The reference webpage has been provided on page 8, but please move it here as this is the first time that EDCNet™ is mentioned.  

Page 5, Materials and Methods, 2.1 Quality assurance programs offered, Paragraph 2

·        There is mention of NRL EQAS including five sample vials of different HCV genotypes. What are these different genotypes? Is there a reason why these are restricted to 5 different genotypes, when there are 8 known genotypes around the world?

Page 7, Materials and Methods, 2.7 Cost

·        It would be useful to provide more details on cost with actual figures (or range of figures) with respect to estimated bulk manufacturing costs and perhaps a comparison of shipping costs between inactivated dried tube samples vs serum/plasma samples.

Page 7, Results, Figure 1

·        I would recommend putting both Genotype 3 and Genotype 1 plots in the Figure 1 panel. If both genotypes have been tested over a period of 15 months, we should see the data for both.

Page 7, Results, 3.1.2 Accelerated stability testing

·        Line 3 – at the end of the line, the new sentence starts with ‘At’, but should start with ‘A’.

Page 8, Results, Figure 2

·        Please do some basic statistics on these results to include medians and interquartile ranges of viral loads at each temperature (this information can be added in the caption or put on the figure itself). Please also do a statistical comparison test to show that there are no significant differences in median viral load at different storage temperatures.

Page 8, Results, 3.2 Data collection

·        Line 3 – please replace ‘build’ with ‘built’ in the sentence, “The program, build within NRL’s EDCNet…..”

·        Line 4 – please move the webpage reference to EDCNet to the first time this software is mentioned on page 5.

Page 10, Discussion, Paragraph 6

·        Line 1 – in the sentence “…and QR codes with simple, initiative data entry….”, do the authors in fact mean ‘intuitive’?
